# Relevance of the Get Active Questionnaire for Pre-Participation Exercise Screening in the General Population in a Tropical Environment

**DOI:** 10.3390/healthcare12080815

**Published:** 2024-04-10

**Authors:** Cuiying Lisa Ho, Venkataraman Anantharaman

**Affiliations:** 1Department of Orthopaedic Surgery, Sengkang General Hospital, 110 Sengkang East Way, Singapore 544886, Singapore; 2Department of Emergency Medicine, Singapore General Hospital, Duke-NUS Academic Medical Centre, Outram Road, Singapore 169608, Singapore

**Keywords:** physical activity, preparticipation, screening

## Abstract

The Get Active Questionnaire (GAQ), developed by the Canadian Society for Exercise Professionals (CSEP), was recently recommended for pre-participation screening of the general population in Singapore before increasing their exercise levels. This literature review examines the evidence behind the GAQ and its relevance to our tropical environment. Searches were carried out via Pubmed, MEDLINE and the Cochrane Central Register of Controlled Trials. Resources referenced by the CSEPs were hand searched. The CSEP was also contacted for further information. The evidence behind each GAQ question was compared to international literature and guidelines, where applicable. Out of 273 studies, 49 were suitable for analysis. Two GAQ studies commissioned by the CSEP showed a high negative predictive value but high false negative rate. Of the nine GAQ questions, those on dizziness, joint pains and chronic diseases appear to be justified. Those on heart disease/stroke, hypertension, breathlessness and concussion require modification. The one on syncope can be amalgamated into the dizziness question. The remaining question may be deleted. No long-term studies were available to validate the use of the GAQ. Heat disorders were not considered in the GAQ. Modification of the GAQ, including the inclusion of environmental factors, may make it more suitable for the general population and should be considered.

## 1. Introduction

Regular physical activity is associated with numerous health benefits such as reducing rates of obesity and cardiovascular disease and improving quality of life [1]. The United Kingdom Chief Medical Officers’ Physical Activity Guidelines recommend 150 min of moderate-intensity aerobic exercise per week, 2 or more days a week of strengthening activity for adults and the elderly and 60 min of moderate to vigorous activity a day for children [2,3]. There are differences in the activity requirements in Asian populations [4].

Despite the extremely low absolute risk of sudden death during vigorous exercise in asymptomatic individuals, this is increased in sedentary individuals [5]. An epidemiological study of 1247 sudden cardiac arrest cases during sporting activity showed that 5% were middle aged (51.1 ± 8.8 years) [6]. In the 12–35-year-old sporting population, competitive sports increased the risk of sudden death 2.5-fold [7]. Hence, a simple screening tool for the general population is needed to identify those at higher risk of adverse consequences during strenuous physical activity.

In 2019, the Singapore Sports Safety Committee recommended the Get Active Questionnaire (GAQ) for pre-participation screening in place of the previously used Physical Activity Readiness Questionnaire (PAR-Q) [8]. The increasing length and complexity of the PAR-Q documents were possible barriers to people starting exercise. The aim of this review was to examine the evidence behind the GAQ and assess its applicability to tropical and certain temperate environments, with their adverse implications on the health of local dwellers and travellers, especially when involved in increased physical activity [9].

The GAQ was developed over a 2-year period by the Canadian Society for Exercise Physiology (CSEP) in 2017. Its aim was to create a self-administered and simple tool for all ages to “screen in” healthy physical activity. CSEP members (academics, students of human kinetics and exercise physiology, industry and allied partners) developed an initial version based on an online member survey and current screening procedures [10]. Three studies were commissioned to evaluate the effectiveness of the GAQ; one each for children, healthy adults and older adults, of which only two were published [11,12]. There was subsequent refinement of the GAQ before it was released. 

The GAQ is a two-page document with two accompanying reference sheets for those answering “Yes” to any question [13,14]. The first page has nine questions (a total of four, with the first one being in six parts) numbered as follows: 1A, 1B, 1C, 1D, 1E, 1F, 2, 3 and 4. The answer of “Yes” to any one of these nine questions will require the use of the two-page reference document for advice on what to do and a likely referral to a medical practitioner or a qualified exercise professional for further evaluation. The answer of “No” to all the nine questions would suggest that the person is free to participate in or increase their level of PA and should move to page 2 of the main GAQ document to assess their current physical activity. The second page assesses current baseline physical activity and provides general guidance on physical activity for healthy living [13]. The reference pages provide advice on the nine questions asked earlier [14]. There is no specific validity period and exercise participants are recommended to re-do the questionnaire if their health changes.

The aim of this review was to examine the evidence behind each question of the GAQ and evaluate its applicability for a tropical environment. 

## 2. Materials and Methods

A literature review was undertaken via the databases PubMed, MED-LINE and the Cochrane Central Register of Controlled Trials Register (1996 through 2023). The keywords and MeSH terms used were get active questionnaire, pre-participation screening, sudden death and physical activity. We also used key words relevant to each of the nine questions in the GAQ to search for studies on these questions in relation to physical activity and clinical outcomes. Relevant reference lists were also hand searched. Studies on conducting pre-participation screening among children and adults were deemed eligible. Studies which were mainly addressing the need for physical activity as therapy for a variety of medical illnesses were excluded. The search was limited to studies in English and on human subjects. Review papers were also included if they covered studies not included in systematic reviews and meta-analyses. If included, these review papers, their reference lists and primary studies were also hand searched. The SCEP website was also searched to obtain the GAQ references used in development of their documents, as well as press releases and a GAQ learning module for health professionals. We wrote to the CSEP to obtain information on the basis for the GAQ. We also reviewed the American and European guidelines and the relevant studies pertaining to each question in the GAQ.

We assessed the papers and studies according to study characteristics (viz. year of publication, study design, number of subjects and age ranges, where applicable) and study methodology (reasons for patient inclusion and exclusion, test outcomes, if any, to determine impact of physical activity, outcomes of interventions carried out, if any, and study attrition for studies conducted over periods of weeks or months). The two authors initially conducted the reviews independently, and subsequently reviewed these together over a series of meetings. This literature review was not registered in any review database owing to this not being a systematic review.

## 3. Results

Of the 273 studies found, 224 were rejected because they used the same research findings as the basis for their recommendations. Forty-nine were found to be suitable and were included in this study. Two papers were on the applicability of the GAQ in different populations. There were 16 consensus statements, 16 systematic reviews and/or meta-analysis and 17 individual studies. 

### 3.1. Published Studies Commissioned by the CSEP

The CSEP recommended the GAQ on the basis of three commissioned studies, two of which were published. One was based on elderly adults of age 75 ± 7 years and the other on children [11,12]. We were unable to obtain any further information from the CSEP.

For a convenience sample of 112 elderly adults in the community, exercise stress testing was used as a gold standard and compared with the participants’ responses on the GAQ [11,15]. All participants responded to the GAQ and underwent graded exercise stress testing (EST) on the treadmill. The participants returned a week later and repeated the GAQ to assess its test–retest reliability. The GAQ performed well as a “screen-in” tool with a high negative predictive value (83.5%) and GAQ and EST agreement of 79.8%. However, the GAQ was not as sensitive as EST in detecting at-risk participants for exercise (positive predictive value = 16.7%) and had a high false negative rate (94.5%). Of the nine GAQ questions, six were deemed to have substantial test-retest agreement (1A, 1B, 1C, 2,3 and 4), one had moderate agreement (1E), one had fair agreement (1D) and one (1F) could not be assessed as only one person answered this question. Furthermore, the frequency of “yes” responses in the GAQ was small and ranged from 1–18 participants (one for 1F on concussion and eighteen for question 2 on joint pains) [11].

The GAQ was also answered by the parents of 207 children (mean age 8.4 years) when they attended emergency medicine and various specialist outpatient clinics (e.g., neurology, respiratory, rehabilitation, chronic pain, rheumatology and cardiology) [11]. The children’s attending doctor and medical records were used as a “gold standard” compared to the parents’ ability to reliably identify factors that would limit their child’s physical activity. The GAQ had a high “screen-in” rate in children, with a high negative predictive value (92%) and a high sensitivity rate (71%), and could pick up children with cardiac or arthritic conditions that were of concern. However, it had low specificity (59%) and a low positive predictive value (23%). This study population included a higher percentage of children with chronic diseases as compared to a general healthy population of children. The low specificity was explained by parents being over-protective and having higher perceived physical activity restrictions as compared to the doctors. The GAQ had a high false negative rate (29%), especially in children with epilepsy who needed supervision while swimming but experienced no symptoms during the period covered by the GAQ. This was not a physical activity restriction but a special consideration when the child engaged in physical activity. Hence, the authors recommended additional questions to pick up special considerations for children exercising, e.g., supervised physical activity and medication adjustments in children with diabetes or asthma [11].

### 3.2. Itemised Evaluation of the Questions in the GAQ


*Evidence behind the GAQ Question 1A: Have you experienced, within the last six months, a diagnosis of/treatment for heart disease or stroke, or pain/discomfort/pressure in your chest during activities of daily living or during physical activity?*


This question aims to screen for cardio-cerebro-vascular and sudden cardiac arrest (SCA) risks during exercise. It is a major component of every pre-participation screening system. Although regular exercise results in long term risk reduction for cardiovascular disease, each session can transiently increase the risk of SCA as evidenced by news reports on apparently healthy children or adults dying while exercising [1]. SCA in individuals under 35 years age can be due to disorders of myocardial structure or conduction. In individuals older than 35 years, it is often owing to atherosclerosis [16,17]. 

The prognostic value of exercise-induced ventricular arrhythmias during exercise testing is uncertain, except in those at high risk of coronary ischemia or with existing myocardial damage, hypertrophic cardiomyopathy or arrhythmogenic right ventricular cardiomyopathy [18,19]. The adverse cardiac event rate for patients with established heart disease on cardiac rehabilitation is approximately double that for the general public [20]. In men older than 70 years the relative risk of a cardiac event during and following exercise at intensity > 6 METs was 12.7 [21]. These studies suggest that the risks of adverse events during physical activity are greater for patients with cardiovascular disease if they are not medically stable and not currently physically active [1]. 

Pre-participation questionnaires are akin to screening by history alone. Although their specificity is high at 94%, their sensitivity is low (20%) as compared to ECG (94%) but better than physical examination (9%) [22]. Although the majority of SCA patients are reportedly asymptomatic until the terminal event, it is an important question to ask as it prompts symptomatic patients to seek medical review or educate those who may not be aware of the warning symptoms [23,24]. 

Patients with increased risk may also present with early symptoms of exercise-associated cardiac symptoms, particularly syncope [25]. The incidence of reported symptoms prior to SCA may be lower owing to insufficient documentation in the medical, paramedical or witness accounts. The Oregon Sudden Unexpected Death Study (Oregon-SUDS) reported that, out of 839 patients with SCA, 51% reported warning symptoms such as typical chest pain (46%), breathlessness (18%), syncope or palpitations (5%) and/or influenza-like symptoms (10%) up to a month before the event [26]. Such patients, if picked up via screening, would have required medical review prior to starting at least moderate physical activity. Since there is reasonable evidence to suggest that patients with prior cardiac disease or cardiac symptoms within six months are at increased risk of further myocardial injury, the inclusion of heart disease and history of chest pain symptoms in question 1A seems reasonable. 

The same cannot be said of the adverse effects of physical exertion in patients with recent stroke. There is insufficient evidence to suggest that exercise after a stroke increases the risk of adverse outcomes [27]. The need for individuals with an acute stroke (<6 months) to seek medical clearance prior to participating in physical activity requires review. The inclusion of stroke in 1A is, presumably, due to the similar risk factors it shares with ischaemic heart disease.


*Evidence behind the GAQ Question 1B: Have you experienced, within the last six months, a diagnosis of/treatment for high blood pressure (BP), or a resting BP of 160/90 mmHg or higher?*


Even though hypertension is generally regarded as a major risk factor for coronary artery disease, there is, to date, no evidence indicating a major risk of adverse cardiac events in hypertensive patients undertaking moderate-to-intense physical activity. A review of 50 papers on exercise interventions in hypertensive subjects did not report adverse events if their baseline systolic blood pressure (SBP) was between 140–180 mm Hg (mid-point 160 mm Hg) and their mean diastolic blood pressure (DBP) was 90 mm Hg [1]. The European Society of Cardiology’s (ESC) 2020 guidelines define poorly controlled hypertension as a resting SBP exceeding 160/90 mm Hg and recommends holding off maximal EST until the patient’s SBP is below this reading [23]. These suggest that hypertensive patients may be regarded as low risk if they are medically stable and their BP < 160/90 mm Hg.

The ACSM, however, does not specify exclusion criteria and any patient with asymptomatic hypertension need not undergo medical clearance before exercise [5]. The high prevalence of cardiovascular risk factors in the United States and Canada can potentially result in 95% of men and women over 40 years age requiring medical clearance before exercise, based on their previous guidelines [28]. Hence their recommendation that physically active asymptomatic individuals with cardiovascular, metabolic or renal disease can continue exercising with progressive intensity unless new symptoms develop. The BP-lowering effects of exercise are indisputable [29]. We presume that the inclusion of 1B in the GAQ is a result of hypertension being generally regarded as a major risk factor for heart disease and to flag up patients with uncontrolled hypertension who may need medical therapy and a supervised exercise programme to start with. It is uncommon for many patients to be aware of their resting blood pressures, making this an honest entry in pre-participation screening.


*Evidence behind the GAQ Question 1C: Have you experienced, within the last six months, dizziness or light-headedness during physical activity?*


Dizziness, especially during or after physical activity, is often seen as a pre-syncopal symptom of arrythmias. The main cardiac conditions associated with malignant arrhythmias and increased SCA risk during exercise are hypertrophic cardiomyopathy, arrhythmogenic right ventricular cardiomyopathy and prolonged QT syndromes [30]. Patients with non-lethal arrhythmias can exercise without increased risk if they are medically stable and engage in regular physical activity for >20 min at least three times a week [1]. Amongst adults, 21.4% cases of SCA following exercise over a 5-year period had herald symptoms like chest pain or syncope [31]. Amongst children who suffered from SCA, 24% had at least one warning event [32]. The top three most common symptoms, viz. recent fatigue (44%), near syncope/light-headedness (30%) and chest pain/palpitations (28%), were reported within an average of 30 months before the event [32]. Therefore, dizziness during physical activity is a symptom to be taken seriously in pre-participation screening.


*Evidence behind the GAQ Question 1D: Have you experienced, within the last six months, shortness of breath at rest?*


Breathlessness is a general symptom that usually refers to cardiovascular and/or respiratory disease. Outcomes of prior cardiovascular disease with physical exercise have been described earlier. There is little risk of acute exacerbation with exercise in patients with well-controlled asthma [33]. However, in patients with chronic obstructive pulmonary disease (COPD), associations with sedentary lifestyle and cardiovascular diseases can increase the risk of adverse events during exercise [3]. In the USA the estimated death rate from asthma amongst competitive athletes is 0.23 per million [34]. Since the absolute increase in asthma death during physical activity is very small, sports should be encouraged in asthmatics [35]. Furthermore, high-intensity exercise training in moderate-to-severe COPD is not only safe, but also beneficial as these patients demonstrate an increase in aerobic capacity after 12 weeks of exercise [36]. 

The Oregon-SUDS study reported an 18% incidence of unusual breathlessness in the four weeks prior to sudden cardiac arrest [26]. Hence, there would be a need to pay attention to any presence of unusual breathlessness during exertion as that may suggest either cardiac, respiratory or other causes (anaemia), all of which may have negative consequences if not investigated early. 


*Evidence behind the GAQ Question 1E: Have you experienced, within the last six months, loss of consciousness/fainting for any reason?*


Syncope at rest or during exertion has many causes (covered in the discussion on question 1C). It may suggest structural heart disease or malignant arrhythmias. About 5% of children presenting with syncope have a cardiac aetiology [37]. This is also noted in non-Caucasian populations [38]. Amongst adults, 33% of athletes who presented with syncope during exercise have structural heart disease, which can cause SCA [39]. Amongst paediatric SCA patients, 24% had, on average, 2.6 episodes of syncope or unexplained seizure activity before the event, with the first episode being as early as 4 years ago [32]. Syncope is an important and valid symptom to screen for potentially malignant rhythms that may be triggered during exercise and prompt further investigations. 


*Evidence behind the GAQ Question 1F: Have you suffered concussion within the last six months?*


Concussion (a transient symptom of traumatic brain injury or TBI) is common among children and adolescents participating in contact sports [40]. Although absolute rest is no longer recommended for patients with concussion, athletes who participate in high-intensity sports following a recent concussion experience more symptoms and have worse neurocognitive performance than those who undergo moderate levels of activity [41]. TBI results in the need for adapted strategies to increase participation in physical activity [42]. Patients with persistent symptoms need a gradual return to active moderate intensity exercise [40]. 


*Evidence behind the GAQ Question 2: Do you currently have pain or swelling in any part of your body (such as from an injury, acute flare-up of arthritis, or back pain) that affects your ability to be physically active?*


The global burden of musculoskeletal diseases is very large [43]. Patients usually present with pain, whether acute or chronic. While acute pain requires rest, recovery from chronic pain requires the patient to continue being physically active despite the pain [44,45]. Joint problems predispose a participant to limited or modified physical activity. Regular low-impact aerobic exercises, strengthening exercises and neuromuscular training can maintain physical function and reduce pain levels in patients with osteoarthritis [46]. The International Olympic Committee (IOC) recommends an element of musculoskeletal screening to detect current injuries because a previous injury is the most consistent risk factor for recurrent injuries [47]. For example, a previous ankle sprain is linked to a higher risk of reinjury and contralateral side injury and may require further rehabilitation before commencing exercise [48].


*Evidence behind the GAQ Question 3: Has a health care provider told you that you should avoid or modify certain types of physical activity?*


Healthcare providers would usually use a patient’s medical condition as a basis to advise on physical activity modifications, which could be covered in Question 4. Other than as a safety-net enquiry, it is difficult to deduce evidence for such a vague question, especially with the lack of explanation for this from the CSEP.


*Evidence behind the GAQ Question 4: Do you have any other medical or physical condition (such as diabetes, cancer, osteoporosis, asthma, spinal cord injury) that may affect your ability to be physically active?*


Asthma was covered in the GAQ 1D. This section addresses the evidence behind the risk of exercise in patients with diabetes, cancer, osteoporosis, spinal cord injury, and other chronic medical conditions.

Risks associated with exercise in diabetics, viz. cardiac events and hypoglycaemia, are for low-to-moderate-intensity physical activity. There is no current evidence to suggest that additional screening is required beyond usual diabetes care to reduce this risk [49]. Patients with diabetes who have end-organ complications may be at higher risk of adverse events with increased physical activity and may benefit from referral to a health care provider for review prior to increasing their exercise intensity [49].

In general, cancer survivors can gradually start low intensity aerobic or resistance training without medical clearance. Pre-participation screening and exercise precautions, in addition to improving common side effects of a cancer diagnosis and treatment, are recommended in those with peripheral neuropathy, lymphedema, cardiopulmonary disease, severe nutritional deficiencies and bone metastases [50]. 

Patients with osteoporosis are at a higher risk of experiencing adverse events during exercise if they have a fragility fracture after the age of 40 years, take systemic corticosteroids for at least three months and have had more than two falls in the previous 12 months [51]. Those with stable osteoporosis can safely carry out aerobic or resistance training as it improves physical function and pain. 

Regular exercise improves exercise capacity and reduces depression in patients with spinal cord injuries [52]. These patients would benefit from pre-participation medical review for multiple complications associated with spinal cord injuries as certain precautions need to be taken during exercise [52].

For other chronic medical illnesses such as renal disease or epilepsy, the possible risks of participating in exercise are adverse events secondary to concomitant cardiovascular and musculoskeletal disease and syncopal episodes secondary to epilepsy [53]. There have been no reports of cardiac arrest or musculoskeletal injury occurring in exercise testing carried out on patients with these illnesses [53,54]. Pre-participation screening can help address appropriate concerns.

### 3.3. Issues Not Addressed in GAQ

The GAQ was designed for a Canadian population. In tropical and some temperate countries, especially in humid conditions, heat disorders occur with strenuous physical activity [22]. These are more likely after mild illnesses such as viral infections and gastroenteritis in the few days prior to intense physical activity. A retrospective review of two large USA-based cardiovascular registries revealed that 7% of sudden deaths were due to myocarditis, of which 60% (58/97) occurred during or just after physical activity [55]. In 46 patients in whom symptoms were recorded, 35% had viral illness, 19% syncope, 15% nausea/abdominal pain and 15% chest pain and palpitations [55]. Heat disorders are of concern in these environments and may need inclusion in pre-participation screening for physical activity.

The GAQ included frequent references to consulting qualified exercise professionals for those wishing to engage in or increase their level of physical activity. Such persons are not easily available in most countries and most will need to see a doctor for such advice [56]. Members of the medical community in most countries may not understand what investigations to carry out or what advice to give.

## 4. Discussion

The CSEP states that “pre-screening for physical activity using an evidence-based screening tool is an important first step in ensuring a safe and enjoyable physical activity experience” and that it is “to easily screen-in the majority of Canadians to safely participate in physical activity and exercise” [57]. For most, the decision should be to proceed with the physical activity. Only a minority should need medical review.

From the results presented the evidence-based status of the GAQ questions appears to be as follows:Question 1A: Since patients with prior cardiac disease, who are not medically stable, physically inactive or have experienced cardiac symptoms within the last six months are at increased risk of further myocardial injury, the inclusion of heart disease and history of chest pain symptoms seems reasonable. There is lack of similar evidence for stroke patients.Question 1B: The current evidence does not justify the inclusion of hypertension based on a set of blood pressure recordings which most participants may not be aware of. It appears reasonable to include non-technical descriptors of symptomatic hypertension in the questionnaire.Question 1C: Dizziness has multiple causes. Some may portend conditions that predispose patients to SCA, especially during exercise. Their inclusion in the questionnaire appears valid.Question 1D: The occurrence of unusual breathlessness during prior physical activity, rather than at rest, appears to be an early warning symptom for adverse consequences during exercise. Those who are already breathless at rest would be highly unlikely to want to engage in increased physical activity.Question 1E: The major overlap of the factors in this question with those in question 1C suggests that both may be combined into a single one.Question 1F: The tendency of TBI patients with persistent post-concussive symptoms to have worsening effects with increasing physical activity necessitates its inclusion in the questionnaire.Question 2: With an IOC recommendation and evidence that persistent joint pains aggravate symptoms with increase in physical activity levels, this question appears reasonable.Question 3: Other than as a safety-net question there is no other likely reason for its continuing inclusion.Question 4: Patients with uncontrolled chronic medical illnesses are more prone to adverse events during physical activity. These conditions need to be identified.

The sample size of the CSEP-commissioned study on elderly subjects was not adequate to address the usefulness of the nine individual questions in the GAQ. For the study commissioned on children, the GAQ had a high false negative rate, low specificity and a low positive predictive value. The study commissioned on the largest group of exercise participants, viz. young and middle-aged adults, was not released. The GAQ does not address the issue of heat disorders that confronts sports participants in many communities. The document promoted a heavy reliance on qualified exercise professionals, a rare commodity in most communities. There are also no reports that validate the use of the GAQ nor demonstrate a decrease in numbers of adverse events in exercise participants following its use.

The GAQ is a laudable attempt at a comprehensive pre-participation screening tool. It is meant to cover all ages and genders and assess exercise risk in patients with cardiovascular and other chronic medical disease. The quantity of questions and overlap of potential risk factors can be a barrier to lay persons completing the questionnaire. There would also be concern about whether the range of conditions highlighted may contribute to restricting exercise in individuals with these issues. Though the GAQ uses some technical technology, such as “dizziness”, “osteoporosis” and “concussion”, it is not clear if these terms may be too complex for individuals with a low level of education. There are no studies analysing this issue, even in Singapore with its very high literacy rate. However, general literacy (the ability to read and write) may not be the same as health literacy (the degree to which individuals have the capacity to obtain, process and understand basic health information and services needed to make appropriate health decisions) and the relevance of this to understanding such health-related questionnaires would be a matter that requires further evaluation.

### Limitations

One of the key limitations of this review was that most of the studies we reviewed were conducted in high-income countries such as Canada, The United States of America, Australia and the countries of Europe, including the United Kingdom. Similar work in the rest of the world has been very limited and is not represented in the published literature.

## 5. Conclusions

While the GAQ is meant to be widely applicable, there is a need to simplify the tool, minimize the amount of duplication in its content and include other relevant areas. Further studies are required to validate how useful the GAQ will be in detecting conditions in the general population that require medical clearance and in the prevention of adverse events without inhibiting active participation in physical activity.

## Data Availability

Not applicable.

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
