# Peer review of "Relevance of the Get Active Questionnaire for Pre-Participation Exercise Screening in the General Population in a Tropical Environment"

_healthcare, 2024, doi:10.3390/healthcare12080815_

Round 1

Reviewer 1 Report

Comments and Suggestions for Authors

Dear all,

This is an motivating study, which aimed to examine the evidence behind each question of the Get Active Questionnaire (GAQ) and evaluate its applicability for a tropical environment.

The study is in line with the purpose of the journal, and the subject matter provides the academics and professionals working in the sector with insightful information.

The layout of the current paper could be done with some adjustments. The authors have done a nice job describing the study.  I have a some comments below:

Title

Adequate

Abstract

The abstract should be a single paragraph, and should follow the style of structured abstracts, but without headings. Background: Place question addressed in a broad context and highlight the purpose of the study; Methods: Describe briefly the main methods or treatments applied; Results: Summarize the article's main findings; and Conclusion: Indicate the main conclusions or interpretations.

The Keywords shouldn’t be repeated from the title “get active questionnaire”.

Introduction

In line 52; Background of the GAQ, please delete.

Materials and Methods

This part has been significantly shortened and reduced in a way that raises a lot of questions from the reader. You should clarify this part and describe with satisfactory detail to let other researchers to replicate and build on published results.

Please describe the materials and methods you conducted to answer the below missing subtitles:

  • The review protocol.
  • The research question structure.
  • Inclusion/exclusion criteria
  • Search Strategy
  • Data Extraction
  • Data Synthesis
  • GAQ questions

It will be better to add a figure to present the systematic review process.

Results

Adequate, nevertheless it will be fine if you could add a table or more to present the description of the included studies, the descriptive characteristics of the relevance estimates.

Discussion

I think you should state one of the key limitations is the fact that most the studies were achieved in high income countries (i.e, Australia, Canada).

References

In journal articles references: note that: Journal Name in italic Year is bold

In Books references: note that: Book Title is italicized

Line 432, Ref 27, journal name should be abbreviated (American Journal of Hypertension; Am J Hypertens).

Healthcare recommends the ACS style guide for references, please follow in all inserted references.

With my best regards,

Reviewer 2 Report

Comments and Suggestions for Authors

Thank you for inviting me to review this work. I have some comments on the study:

MAJOR CONCERN:

·      In Question 2, it is crucial to differentiate between acute and chronic persistent pain. Chronic pain conditions, such as fibromyalgia, often involve systemic and psychosocial factors rather than a clear anatomical issue. Despite the pain, moderate-intensity or high-intensity exercise under supervision may be recommended in these cases. With the global prevalence of musculoskeletal pain exceeding 80%, it is evident that nearly everyone will experience musculoskeletal pain at some point in their lives (see the Lancet series papers on musculoskeletal pain and the global burden of disease). Therefore, it is important to distinguish between acute stages, where the recovery process may still be underway, and chronic pain conditions, where individuals must learn to remain physically active despite the pain.

MINOR CONCERNS:

·      [Title] Consider adding "as a screening tool for the general population before increasing exercise levels".

·      [Line 37] Why are only children mentioned in the sentence justified by references 2 and 3?

·      [Line 32-38] Why only reference the American Cancer Society (ACS) recommendations and not those from the World Health Organization (WHO)?

·      [Line 50-51] Please provide more information on what constitutes a tropical environment and its implications.

·      [Line 52-69] The scoring system of the GAQ is unclear. Regarding the PAR-Q, what action should be taken if a person answers 'Yes' to one of the items? Are there different levels of risk stratification?

·      I[Line 52-69] t is unclear why the introduction does not describe the questions by numbers but refers to the page number. Additionally, in the results, the authors use a different system to mention the questions, making it difficult to understand some of the content.

·      [Line 72-73] Please provide each database with a specific search strategy as supplementary material with the specific number of elements in each of them.

·      [Line 70-79] Include the number of evaluators involved in searching and filtering articles in the methods section.

·      [Line 70-79] Explain in the methods section why the review was not registered in any review database.

·      [Line 81] Please be more precise in explaining the reasons for excluding articles rather than just stating they were 'repetitive or not relevant'.

·      [Line 80-84] Consider including a flowchart summarizing the search process.

·      [Line 85-308] A table synthesizing the results of 49 selected studies, along with the references (maybe as supplementary material), would aid the reader's understanding.

·      [Line 309] The questionnaire uses technical terminology that may be complex for individuals with a low level of education (e.g., 'dizziness'). Are there any studies analyzing this issue, specifically in the Singaporean population?

·      [Line 309] Since the introduction mentions that the GAQ is intended to replace the PAR-Q, it would be beneficial for the discussion to compare the two questionnaires and discuss their potential advantages or disadvantages.

Round 2

Reviewer 1 Report

Comments and Suggestions for Authors

Dear all,

Thank you for considering my comments.

The abstract structure still needs to be amended, please delete the words Background, Methods, Results, and Conclusions from the abstract.

The article covers an important topic, nonetheless, it presents outcomes of a systematic review and meta-analysis. I know that including a comprehensive explanation of the materials and methods may help academics to use and build on published works.

In lines 91 and 92; please

Avoid starting a sentence with a numeral. Either write the number in words or rearrange your sentence, For example:

Line 91: “Forty-nine studies were found to be ……..’’

Line 92: ‘’Two papers were on the applicability ……..’’

All the best,

Reviewer 2 Report

Comments and Suggestions for Authors

I have no further comments on the new version of the manuscript. The authors have solved all my concerns with their response.

Author Response

Reviewer 2:

I have no further comments on the new version of the manuscript.

The authors have solved all my concerns with their response.

Thank you.

Please let us know the outcome of our request for publication.

Yours faithfully,

Prof Venkataraman Anantharaman

Department of Emergency Medicine

Singapore General Hospital